# Anisotropic Mechanical Properties of Orthorhombic SiP_2_ Monolayer: A First-Principles Study

**DOI:** 10.3390/molecules28186514

**Published:** 2023-09-08

**Authors:** Yinlong Hou, Kai Ren, Yu Wei, Dan Yang, Zhen Cui, Ke Wang

**Affiliations:** 1School of Automation, Xi’an University of Posts & Telecommunications, Xi’an 710121, China; 2School of Mechanical and Electronic Engineering, Nanjing Forestry University, Nanjing 210042, China; 3School of Automation and Information Engineering, Xi’an University of Technology, Xi’an 710048, China

**Keywords:** SiP_2_ monolayer, first-principles calculations, in-plane anisotropy, Young’s modulus, negative Poisson’s ratio

## Abstract

In recent years, the two-dimensional (2D) orthorhombic SiP_2_ flake has been peeled off successfully by micromechanical exfoliation and it exhibits an excellent performance in photodetection. In this paper, we investigated the mechanical properties and the origin of its anisotropy in an orthorhombic SiP_2_ monolayer through first-principles calculations, which can provide a theoretical basis for utilizing and tailoring the physical properties of a 2D orthorhombic SiP_2_ in the future. We found that the Young’s modulus is up to 113.36 N/m along the *a* direction, while the smallest value is only 17.46 N/m in the *b* direction. The in-plane anisotropic ratio is calculated as 6.49, while a similar anisotropic ratio (~6.55) can also be observed in Poisson’s ratio. Meanwhile, the in-plane anisotropic ratio for the fracture stress of the orthorhombic SiP_2_ monolayer is up to 9.2. These in-plane anisotropic ratios are much larger than in black phosphorus, ReS_2_, and biphenylene. To explain the origin of strong in-plane anisotropy, the interatomic force constants were obtained using the finite-displacement method. It was found that the maximum of interatomic force constant along the *a* direction is 5.79 times of that in the *b* direction, which should be considered as the main origin of the in-plane anisotropy in the orthorhombic SiP_2_ monolayer. In addition, we also found some negative Poisson’s ratios in certain specific orientations, allowing the orthorhombic SiP_2_ monolayer to be applied in next-generation nanomechanics and nanoelectronics.

## 1. Introduction

A material’s physical properties and interaction with external stimuli depend heavily on its geometric structure and symmetry, and the reduced symmetry of materials can bring many unconventional properties and phenomena, such as piezoelectricity [1,2,3], superconductivity [4,5,6], and anisotropy [7,8]. Generally, the reduction in crystal symmetry can be realized by reducing dimensions [9] and surface functionalization [10,11]. Two-dimensional (2D) materials emerge when the thickness of traditional bulk materials decreases to an atomic thickness, which brings numerous intriguing properties. Graphene with a hexagonal lattice is the first successfully prepared 2D material with a high conductivity [12], superior thermal conductivity [13], and large fracture toughness [14]. However, there is no anisotropy in few-layer graphene due to the high-symmetry crystal structure, resulting in the in-plane isotropic electronic, thermal and mechanical properties in 2D graphene. In the following years, various 2D materials have been peeled off, including MoS_2_ [15], h-BN [16], black phosphorene [17], TiS_3_ [18], and biphenylene [19,20]. Among these 2D materials, the lattices of both MoS_2_ and h-BN are hexagonal; thus, their physical properties also are in-plane isotropic. The lattices of black phosphorene, TiS_3_, and biphenylene are rectangle, and they have distinct responses when external stimuli are applied along different crystalline orientations [21,22,23], desirable in potential stress-nanosized sensors and photoelectric devices. For instance, Ling et al. [21] observed the anisotropic responses of black phosphorus to lasers by the angle-resolved absorption and Raman spectroscopies and highlighted the impacts of anisotropy in the electron–photon and electron–phonon interactions. Yarmohammadi et al. [22] tailored the band structure of the black phosphorene monolayer using uniaxial and biaxial strains, and found black phosphorene undergoing the semiconductor-to-insulator and semiconductor-to-semimetal phase transitions under biaxial strains, while keeping the semiconducting phase in the presence of uniaxial strain. Furthermore, it has been reported that the anisotropic materials have a great application potential in aerospace [24], tissue engineering [25], and soft robotics [26,27]. Thus, many researchers have devoted large efforts to designing and preparing anisotropic materials in recent years.

The emergence of 2D orthorhombic SiP_2_ brings a new opportunity for the study of anisotropic materials, due to its rectangular lattice and unique crystal structure. In 2018, Du et al. [28] first explored the electronic and optical properties of 2D orthorhombic silicon diphosphide (SiP_2_) via first-principles calculations. Subsequently, He et al. [29] calculated the anisotropic piezoelectric coefficients in a SiP_2_ monolayer, while Shojaei et al. [30] reported the highly direction-dependent carrier mobility in the SiP_2_ monolayer, yielding the effective separation of photogenerated charge carriers. In 2021, Zhai et al. [31] successfully prepared 2D layered orthorhombic SiP_2_ and fabricated a SiP_2_-based photodetector, and the in-plane anisotropic ratio and dichroic ratio of this SiP_2_-based photodetector were measured as 2 and 1.6, respectively. Recently, Tang et al. [32] discovered an excitonic state with a phonon sideband in 2D orthorhombic SiP_2_ by combining optical reflection measurements and first-principles calculations, where the bound electrons were confined within a one-dimensional P-P chain, whereas holes extended in the 2D orthorhombic SiP_2_ plane. Cheng et al. [33] experimentally demonstrated the effective manipulation of Raman and photoluminescence spectra by the uniaxial strain in 2D orthorhombic SiP_2_, while Zhang et al. [34] calculated a lattice thermal conductivity of 16.23 and 2.22 W/mK along the *a* and *b* directions in 2D orthorhombic SiP_2_ by solving the phonon Boltzmann equation, respectively. In 2023, Yang et al. [35] designed a field-effect transistor based on 2D orthorhombic SiP_2_, and the saturation current reached 10^4^ along the one-dimensional P-P chains, far exceeding the International Technology Roadmap for Semiconductors (ITRS) standards and paving a route for a SiP_2_ field-effect transistor based on its strong anisotropy in the next-generation electronic devices.

Although the strong anisotropy of electronic, thermal, and optical properties in 2D orthorhombic SiP_2_ has been reported, the anisotropic mechanical properties and the origin of anisotropy still need to be explored, which is significant in manipulating and utilizing the physical properties of 2D SiP_2_ in next-generation nanoelectronic devices and soft robotics. As is known, it is necessary to consider the mechanical anisotropy of materials to compensate for the intolerance brittle of inorganic materials [36] when designing bionic composites, such as those in teeth and bones. Furthermore, compared with MoS_2_ and black phosphorus, researchers still know little about the 2D orthorhombic SiP_2_. Therefore, we employed first-principles calculations to study the in-plane anisotropic mechanical properties of a SiP_2_ monolayer. Bulk SiP_2_ has three allotropes including the pyrite-type phase, orthorhombic phase, and tetragonal phase. The pyrite-type phase belongs to the Pa3 (No. 205) space group with non-Van der Waals crystal structure [37], while the orthorhombic and tetragonal phases show a 2D layered structure with Pbam (No. 55) [38] and P-421m (No. 113) [39,40,41] space groups, respectively. The 2D orthorhombic flakes have been prepared [31]; thus, we mainly focused on the 2D SiP_2_ monolayer exfoliated from the orthorhombic phase in this paper. We found that the in-plane anisotropic ratios are up to 6.49 and 6.55 for the Young’s moduli and Poisson’s ratio, respectively. In addition, we also discussed the origin of anisotropy in 2D orthorhombic SiP_2_ by analyzing the interatomic force constants. These results can help deepen our understanding of 2D orthorhombic SiP_2_ and provide a theoretical basis for the applications of 2D orthorhombic SiP_2_ in novel nanoelectronic devices and stress-nanosized sensors.

## 2. Results and Discussion

### 2.1. Geometrical Structure and Stability

The top and side views of an orthorhombic SiP_2_ monolayer are presented in Figure 1, where the blue dashed line marks the unit cell. In Figure 1, the dark green and red balls symbolize the Si and P atoms, respectively. The orthorhombic SiP_2_ monolayer belongs to the Pmc21 (No. 26) space group, composed of twelve atoms including eight P atoms and four Si atoms. After establishing the orthorhombic SiP_2_ monolayer model, we optimized its lattice constants (*a* and *b*) independently due to its rectangular lattice. The calculated results are shown in Figure 2a,b. It can be found from Figure 2a that the energy of the orthorhombic SiP_2_ monolayer reached the minimum value (−66.437 eV) as *a* = 3.46 Å. Then, the lattice constant *b* was optimized when *a* was set as 3.46 Å. In Figure 2b, the energy of the orthorhombic SiP_2_ monolayer reached the minimum value (−66.445 eV) when *b* = 10.28 Å. According to the lowest energy principle, the optimized lattice constants of the orthorhombic SiP_2_ monolayer are *a* = 3.46 Å and *b* = 10.28 Å, which is slightly larger than the experimental result of the bulk orthorhombic SiP_2_ [37] but consistent with previous theoretical results [28,29,30]. Afterward, the atomic positions were relaxed in the optimized unit cell until the energy and Hellmann–Feynman force reach the convergence limits. In Figure 2c, the high-symmetry path in the irreducible Brillouin zone is plotted, which would be used in the calculations of phonon dispersion and electronic band structures.

To identify the stability of the relaxed orthorhombic SiP_2_ monolayer, phonon dispersion and AIMD simulation were performed and the obtained results are presented in Figure 3a,b, respectively. In Figure 3a, there are 36 branches including 33 optical and 3 acoustic branches, due to 12 atoms in the unit cell of the orthorhombic SiP_2_ monolayer. Meanwhile, the lowest imaginary frequency is about ~−0.0019 THz at the center of the irreducible Brillouin zone. This slight imaginary frequency indicates that the orthorhombic SiP_2_ monolayer is unstable to the long-wavelength distortion, which has also been found in other 2D materials [42,43,44]. The highest phonon frequency is up to 15.66 THz, revealing the good dynamic stability of our relaxed orthorhombic SiP_2_ monolayer. In Figure 3b, it can be found from the inset that there is no structural distortion, bond breaking, or phase transition observable in the orthorhombic SiP_2_ monolayer after 5000-fs AIMD simulation at 300 K, indicating the robust thermodynamic stability of the optimized orthorhombic SiP_2_ monolayer at room temperature. Furthermore, the fluctuations of total energy for the 5 × 2 × 1 orthorhombic SiP_2_ supercell shows a convergence, also identifying the thermodynamic stability of the optimized orthorhombic SiP_2_ monolayer.

### 2.2. Anisotropic Mechanical Properties

The elastic constants were calculated as $C_{11}$ = 115.59 N/m, $C_{12}$ = $C_{21}$ = 6.30 N/m, $C_{22}$ = 17.81 N/m, and $C_{66}$ = 28.5923 N/m for the orthorhombic SiP_2_ monolayer. These results satisfy the Born–Huang stability criteria [45], revealing the mechanical stability of our optimized orthorhombic SiP_2_ monolayer. Subsequently, we estimated the Young’s modulus (*E*(*θ*)) and Poisson’s ratio (*ν*(*θ*)) by [46,47,48]:(1)$$E\left(\theta \right)=\frac{C_{11}C_{22}-C_{12}C_{21}}{C_{11}\sin^{4}\theta +[\left(C_{11}C_{22}-C_{12}C_{21}\right)/C_{66}-2C_{12}]\sin^{2}\theta \cos^{2}\theta +C_{22}\cos^{4}\theta },$$
(2)$$\nu \left(\theta \right)=\frac{C_{12}\sin^{4}\theta -[C_{11}+C_{22}-\left(C_{11}C_{22}-C_{12}C_{21}\right)/C_{66}]\sin^{2}\theta \cos^{2}\theta +C_{12}\cos^{4}\theta }{C_{11}\sin^{4}\theta +[\left(C_{11}C_{22}-C_{12}C_{21}\right)/C_{66}-2C_{12}]\sin^{2}\theta \cos^{2}\theta +C_{22}\cos^{4}\theta }.$$

The obtained angle-dependent 2D Young’s moduli (*E*(*θ*)) and Poisson’s ratio (*ν*(*θ*)) are presented in Figure 4a,b. Obviously, the Young’s moduli of the orthorhombic SiP_2_ monolayer are anisotropic. The largest value of Young’s modulus is 113.36 N/m along the *a* direction (*E*(0)), while the smallest value is 17.46 N/m in the *b* direction (*E*(90)). The largest Young’s modulus of the orthorhombic SiP_2_ monolayer is much larger than black phosphorene (86 N/m) [47], but slightly smaller than MoS_2_ (123 N/m) [49]. Meanwhile, the largest Poisson’s ratio of 0.354 also occurs in the *a* direction, while the Poisson’s ratio in the *b* direction is only 0.054. Interestingly, there are some negative Poisson’s ratios. To observe these negative Poisson’s ratios clearly, the Poisson’s ratios were redrawn in a rectangular coordinate system, where the horizontal axis represents the angle, and the vertical axis represents the value of Poisson’s ratio, as shown in Figure 4c. According to Figure 4c, we can find that the smallest negative Poisson’s ratio is −0.305 along the ~36°, 144°, 216°, and 324° directions. A negative Poisson’s ratio denotes that when a material is subjected to a tensile strain within its elastic range along a specific direction, an expansion would occur in the vertical direction of the applied strain. Otherwise, a contraction would be observed in the corresponding transverse direction, as it was subjected to a compression strain in the specific direction. Furthermore, a larger absolute value of the negative Poisson’s ratio leads to a larger deformation in the vertical direction of applied strain. In the orthorhombic SiP_2_ monolayer, these negative Poisson’s ratios allow its potential applications in next-generation nanomechanics and nanoelectronics, such as superior dampers and nanoauxetic materials [50,51]. These negative Poisson’s ratios should be attributed to the puckered structure of the orthorhombic SiP_2_ monolayer, and similar negative Poisson’s ratios have also been found in black phosphorus [52,53], BP_5_ [54], and SiP_2_ made up of septilateral rings and triangles [55]. To estimate the in-plane anisotropy of the orthorhombic SiP_2_ monolayer quantitatively, the in-plane anisotropic ratio between the *a* and *b* directions was defined as ${\mathrm{PV}}_{a}/{\mathrm{PV}}_{b}$, where PV represents physical values including the Young’s moduli, Poisson’s ratio, and interatomic force constant. Hence, the in-plane anisotropic ratio of Young’s moduli was calculated as 6.49, while that for Poisson’s ratio was 6.55. These in-plane anisotropic ratios are much larger than that in black phosphorus (~3.77) [47], ReS_2_ (~1.58) [56], and biphenylene (~1.22) [57]. These large in-plane anisotropic ratios of the orthorhombic SiP_2_ monolayer indicate a quasi-one-dimensional mechanical behavior along the *a* direction.

The stress–strain relations of the orthorhombic SiP_2_ monolayer under uniaxial and biaxial strains are presented in Figure 5. In Figure 5a, one can find that the fracture strength is 15.02 N/m under a 17% strain along the *a* direction, while the fracture strength is only 1.63 N/m under a 40% uniaxial strain in the *b* direction. The fracture strength of the orthorhombic SiP_2_ monolayer in the *a* direction is larger than black phosphorene (10 N/m), but that in the *b* direction is much smaller than black phosphorene (4 N/m) [58]. The in-plane anisotropic ratio of fracture strength is up to 9.2 for the orthorhombic SiP_2_ monolayer, suggesting a quasi-one-dimensional mechanical behavior of the orthorhombic SiP_2_ monolayer in the *a* direction. More interestingly, the fracture strain along the *b* direction (40%) is more than twice that in the *a* direction (17%). Meanwhile, a hardening phenomenon can be observed in Figure 5b, as the uniaxial strain in the *b* direction exceeds 20%, as marked by the blue dashed square. A similar phenomenon can be also discovered in Figure 5c, when the biaxial strain goes beyond 15%. In Figure 5c, there is also another interesting phenomenon that the steep decrease in stress is interrupted at the strain of 30.4%, and then a fluctuation of stress can be observed within the range of 30.4~40.3%.

To explain these hardening and fluctuation phenomena of the stress–strain relations, we show the geometrical structures of the orthorhombic SiP_2_ monolayer under specific uniaxial and biaxial strains in Figure 6. Meanwhile, we also list the geometrical parameters in Table 1, such as the lattice constant, layer thickness, and bond lengths, which is convenient to observe the lattice deformations under uniaxial and biaxial strains quantitatively. According to these results in Table 1, it can be found that both uniaxial and biaxial tensile strains decrease the layer thickness of the orthorhombic SiP_2_ monolayer. When the tensile strain is applied in the *a* direction, the layer thickness of the SiP_2_ monolayer decreases slightly (from 5.58 Å to 5.43 Å), and the corresponding reduced ratio is only ~2.6%. In this case, both the lengths of P-P and P-Si bonds increase obviously from 2.269 Å and 2.281 Å to 2.369 Å and 2.313 Å, accompanied by the changing of the space group from Pmc21 (No. 26) to PC (No. 7). As the strain is applied along the *b* direction, the layer thickness can be decreased to 4.65 Å with the reduced ratio of ~16.6% before the breaking of the intermediate P-Si bond in the unit cell of the orthorhombic SiP_2_ monolayer, because the strain in the *b* direction flattens the P-P and P-Si bond effectively, as shown in Figure 6b. Moreover, the length of the P-P bond is almost unchanged while the length of the P-Si bond increases to 2.326 Å under 20% strain and 2.362 Å under 40% strain, revealing that it is more robust to the strain in the *b* direction than the P-Si bond. This is because the zigzag P-P chain only arranges along the *a* direction, whereas the P-Si bonds distribute in the whole 2D plane, as shown in Figure 2. Once the biaxial strain is applied, the layer thickness of the orthorhombic SiP_2_ monolayer reduces significantly from 5.58 Å to 4.85 Å under 10.6% strain and 2.76 Å under 30.4% strain. When the biaxial strain reaches 35.4%, the layer thickness is only 1.79 Å, and the reduced ratio is up to ~67.9%. Under the biaxial strain, the P-Si bonds flatten obviously, resulting in this remarkable decrease in layer thickness. Furthermore, a novel geometrical structure composed of four- and six-membered rings can be discovered when the biaxial strain reaches 30.4%, which leads to the fluctuation of stress within the range of 30.4~40.3%. When the biaxial strain reaches 40.1%, the intermediate P-Si bond in the orthorhombic SiP_2_ monolayer breaks and the crystal structure collapses. 

Under the uniaxial strain along the *a* direction, a lattice distortion from orthorhombic to monoclinic can be observed in Figure 6a, while there is no obvious distortion and deformation in the strained orthorhombic SiP_2_ monolayer along the *b* direction until the breakage of the intermediate P-Si bond, as shown in Figure 6b. In Figure 6c, we found that the crystal structure of the orthorhombic SiP_2_ monolayer under a biaxial strain of 30.4% becomes quite different from the pristine and other strained SiP_2_ monolayer, and no bond breakage and distortion can be observed in this new crystal structure. However, we must ask: what are the electrical properties of this new structure? Hence, we calculated the electron density distribution and band structure of this new crystal structure to observe the influence of strain on the electronic properties of the orthorhombic SiP_2_ monolayer. The calculated electron density distributions and band structures are shown in Figure 7. Similar to the electron density distribution of the pristine orthorhombic SiP_2_ monolayer in Figure 7a, the electron density distributes along the quasi-one-dimensional P-P chain in this new crystal structure of the SiP_2_ monolayer, as shown in Figure 7c. In Figure 7a,c, the electron density is separated by the isosurface of 0.07 e/Å^3^. Meanwhile, the calculated band structure of this novel geometrical structure of the SiP_2_ monolayer is presented in Figure 7d, while Figure 7b shows the band structure of the pristine orthorhombic SiP_2_ monolayer as benchmark. In Figure 7b, a band gap of 1.51 eV between the valence band maximum (VBM) and conduction band minimum (CBM) can be noted, suggesting a semiconducting state. Under a 30.4% biaxial strain, a metallic state is introduced, revealing the high sensitivity of the electronic properties of the SiP_2_ monolayer to geometrical structure and strain. This result indicates that the orthorhombic SiP_2_ monolayer can be used to design and fabricate novel stress-nanosized sensors in the future.

### 2.3. Origin of Mechanical In-Plane Anisotropy

To explore the origin of in-plane mechanical anisotropy in the orthorhombic SiP_2_ monolayer, the interatomic force constants are obtained by the finite-displacement method. The interatomic force constant refers to the magnitude of the force that exists between two atoms. The mechanical, thermal, and phononic properties of 2D materials are heavily dependent on the strength of atomic interaction. Furthermore, Zhou et al. [59] have interpreted the quasi-one-dimensional thermal behavior in a borophene monolayer by comparing the interatomic force constant. Therefore, we hope to interpret the origin of the mechanical anisotropy in the orthorhombic SiP_2_ monolayer using the interatomic force constants. In Figure 8a, we present the overview of the interatomic force constants between the nearest-neighboring atoms along each direction, while the largest interatomic force constants along the *a* and *b* directions are plotted in Figure 8b. The strength of the interatomic force constant is represented by the color and width of the lines. For instance, the interatomic force constant with a strength of 8.446 eV/Å is represented by the purple solid line with a width of 5 pounds. Generally, a denser electron distribution contributes to a stronger interatomic force constant. However, in the orthorhombic SiP_2_ monolayer, the maximum interatomic force constant (8.446 eV/Å) occurs between the nearest-neighboring P-Si atoms with the bond length of 2.281 Å. This interatomic force constant between P-Si atoms is slightly larger than that between the nearest-neighboring P-P pairs (6.992 eV/Å) with the bond length of 2.269 Å, although the electron density distributes along the quasi-one-dimensional zigzag P-P chain in Figure 7a. Thus, the fracture strain along the *a* direction is much smaller than the *b* direction, because the quasi-one-dimensional zigzag P-P chain along the *a* direction is more prone to break than the P-Si bond. Furthermore, the length of the P-P bond is more sensitive to strain than the P-Si bond in the orthorhombic SiP_2_ monolayer, as shown in Table 1. It should be noted that both the bonds between the nearest neighboring P-Si and P-P pairs are buckling. Thus, the interatomic force constants between the nearest-neighboring atoms along the *a* and *b* directions are required to explain the origin of in-plane anisotropy. We plotted the largest interatomic force constants along the *a* and b directions in Figure 8b, which are supposed to dominate the mechanical, thermal, and phononic properties along the *a* and *b* directions. In the *a* direction, the maximum strength of interatomic force constant is 1.274 eV/Å, which is between Si atoms with the distance of 3.460 Å. In the *b* direction, the maximum strength of the interatomic force constant is 0.220 eV/Å, which is between P atoms with the distance of 4.238 Å. Obviously, the largest interatomic force constant along the *a* direction is 5.79 times that in the *b* direction. This large in-plane anisotropic ratio of interatomic force constant can be regarded as the most dominant origin of the in-plane mechanical anisotropy (~6.5) in the orthorhombic SiP_2_ monolayer.

## 3. Computational Details 

All of the first-principles calculations were performed in the *Vienna* ab initio *Simulation Package* (VASP5.4) [60,61] with the projected augmented wave (PAW) method. In the orthorhombic SiP_2_ monolayer model, we imposed a 20 Å vacuum space along the *c*-axis to eliminate the interlayer non-physical interaction. After building the monolayer model, the atomic position in the orthorhombic SiP_2_ monolayer was relaxed, and the convergence criteria for the energy and the Hellmann–Feynman force were set as 10^−^^8^ eV and 10^−^^4^ eV/Å, respectively. A 7 × 2 × 1 Monkhorst–Pack (MP) grid was used to sample the irreducible Brillouin zone during the structural relaxations, while a 9 × 9 × 1 MP grid was employed for the self-consistent calculation. To calculate the phonon dispersion of the orthorhombic SiP_2_ monolayer, the PHONOPY code was used to build the 3 × 1 × 1 orthorhombic SiP_2_ supercell and diagonalize the interatomic force constant matrix [62]. To identify the thermodynamic stability, ab initio molecular dynamics (AIMD) simulations with a Nosé–Hoover thermostat were performed for a 5 × 2 × 1 SiP_2_ supercell [63]. In all of the first-principles calculations, we used the Perdew–Burke–Ernzerhof (PBE) of the generalized gradient approximation (GGA) as the exchange-correlation function [64,65]. In the first-principles calculations, a larger cutoff energy leads to a higher calculation accuracy and requires a larger computing resource. For a proper cutoff energy, we tested the influence of the cutoff energy on the total energy of the orthorhombic SiP_2_ monolayer, as shown in Figure 9. It can be found from Figure 1 that the energy decreases obviously with the cut-off energy increasing from 200 eV to 350 eV, and it begins to converge when the cutoff energy reaches 400 eV. To balance the calculation accuracy and required resource, we chose 600 eV as the cutoff energy in all of our calculations.

## 4. Conclusions

In this paper, we have performed a first-principles study to explore the mechanical properties of the orthorhombic SiP_2_ monolayer, which can provide a theoretical basis for the applications of a 2D orthorhombic SiP_2_ in next-generation flexible devices. According to our calculations, we found that the largest Young’s modulus of the orthorhombic SiP_2_ monolayer is up to 113.36 N/m along the *a* direction, while the smallest Young’s modulus is only 17.46 N/m in the *b* direction. The Poisson’s ratios are 0.354 in the *a* direction and 0.054 in the *b* direction, respectively. Meanwhile, the fracture strength is 15.02 N/m under a 17% strain along the *a* direction, while the fracture strength is only 1.63 N/m under a 40% uniaxial strain in the *b* direction. These values reveal a strong in-plane anisotropy in the orthorhombic SiP_2_ monolayer. To estimate the in-plane anisotropy quantitatively, we defined the in-plane anisotropic ratio and calculated the in-plane anisotropic ratios for the Young’s modulus and Poisson’s ratio as 6.49 and 6.55, respectively; these values are much larger than in other anisotropic 2D crystals, including black phosphorus, ReS_2_, and biphenylene. These large in-plane anisotropic ratios suggest a quasi-one-dimensional mechanical behavior along the *a* direction in the orthorhombic SiP_2_ monolayer. To explore the origin of the intriguing mechanical anisotropy in the orthorhombic SiP_2_ monolayer, the interatomic force constants was obtained through the finite-displacement method. Then, we found that the maximum value of the interatomic force constant along the *a* direction is ~5.79 times of that in the *b* direction, which should be recognized as the main origin of the mechanical anisotropy in the orthorhombic SiP_2_ monolayer. These results are significant to the manipulation and utilization of the physical properties of a 2D SiP_2_ in next-generation nanoelectronic devices and soft robotics.

## Figures and Tables

**Figure 1 molecules-28-06514-f001:**
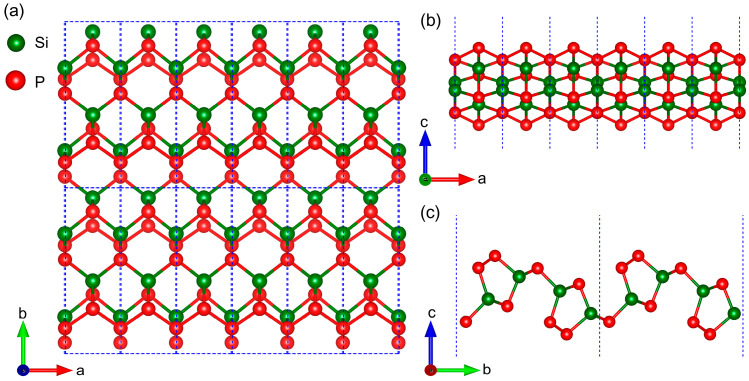
Top (**a**) and side (**b**,**c**) views of an orthorhombic SiP_2_ monolayer. The red and dark green balls represent the P and Si atoms, respectively, while the blue dashed line marks the unit cell of the SiP_2_ monolayer.

**Figure 2 molecules-28-06514-f002:**
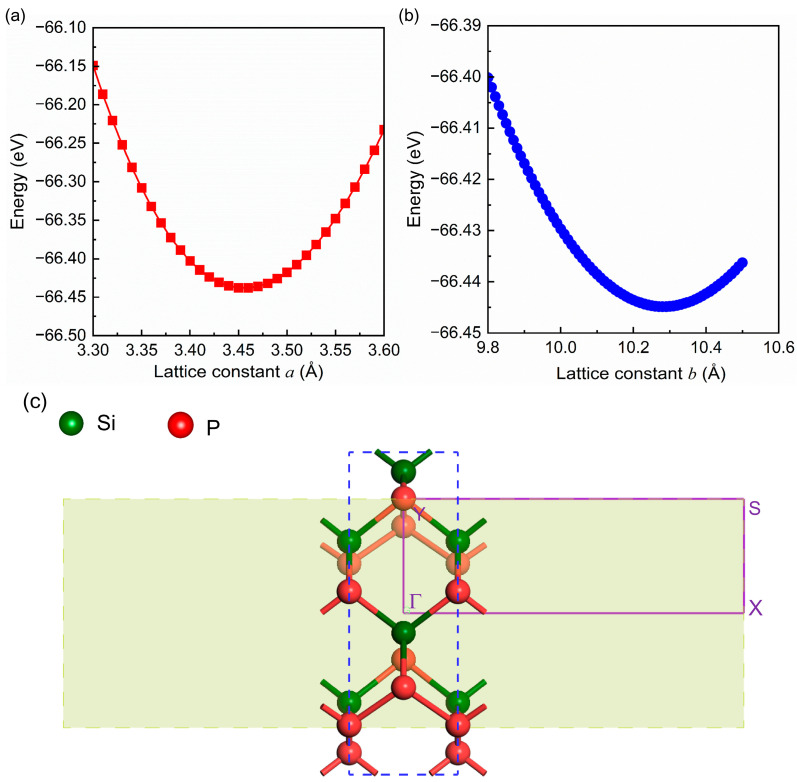
The relationship between total energy and lattice constants *a* (**a**) and *b* (**b**), and the high-symmetry path in the Brillouin zone (**c**). In (**c**), the olive green shadow represents the Brillouin zone, and the purple solid lines are the high-symmetry path. In (**c**), the red and dark green balls represent the P and Si atoms, respectively.

**Figure 3 molecules-28-06514-f003:**
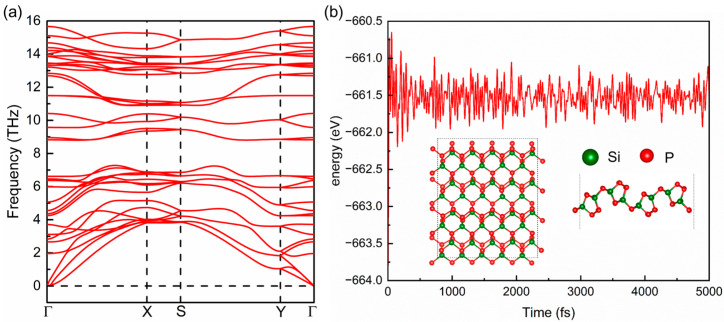
The phonon dispersion of a 3 × 1 × 1 orthorhombic SiP_2_ supercell (**a**), and results of ab initio molecular dynamics (AIMD) simulation at 300 K (**b**). The inset in (**b**) presents the crystal structure of an orthorhombic SiP_2_ supercell after 5000-fs AIMD simulation at 300 K. In (**b**), the red and dark green balls represent the P and Si atoms, respectively.

**Figure 4 molecules-28-06514-f004:**
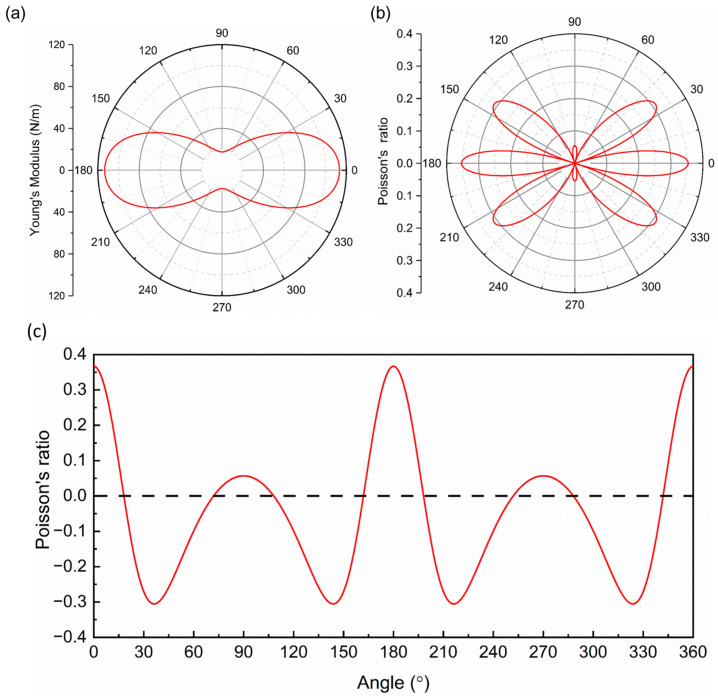
Young’s modulus (**a**) and Poisson’s ratio (**b**) of the orthorhombic SiP_2_ monolayer. (**c**) Poisson’s ratio in a rectangular coordinate system with the horizontal and vertical axes of angle and the value of Poisson’s ratio. The 0 degree in Figure 5a–c represents the *a* direction, while 90 degree represents the *b* direction.

**Figure 5 molecules-28-06514-f005:**
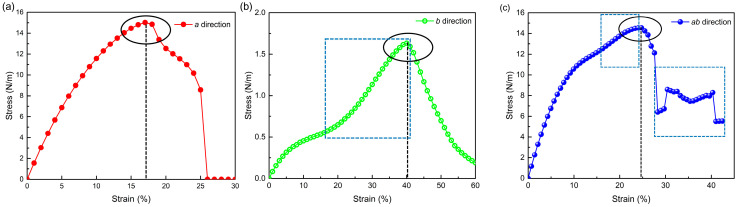
Stress–strain relations under uniaxial (along *a* (**a**) and *b* (**b**) directions) and biaxial (**c**) strains. The black ellipse and black dashed line mark the fracture strength and fracture strain under these uniaxial and biaxial strains, respectively. The blue dashed square marks the interesting hardening and fluctuation phenomena of the stress–strain relations under uniaxial and biaxial strains.

**Figure 6 molecules-28-06514-f006:**
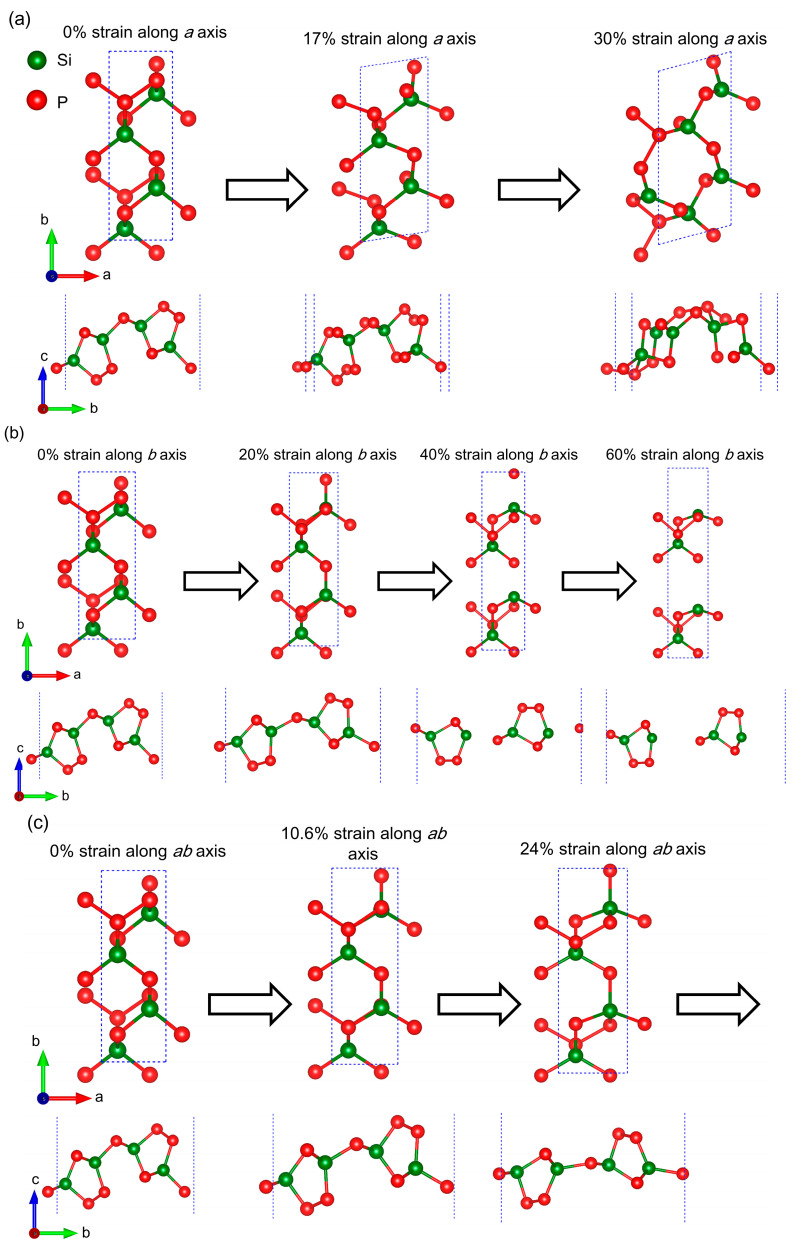
Top and side views of the orthorhombic SiP_2_ monolayer under specific uniaxial (along the *a* (**a**) or *b* (**b**) direction) and biaxial (**c**) strains. The red and dark green balls represent the P and Si atoms, respectively.

**Figure 7 molecules-28-06514-f007:**
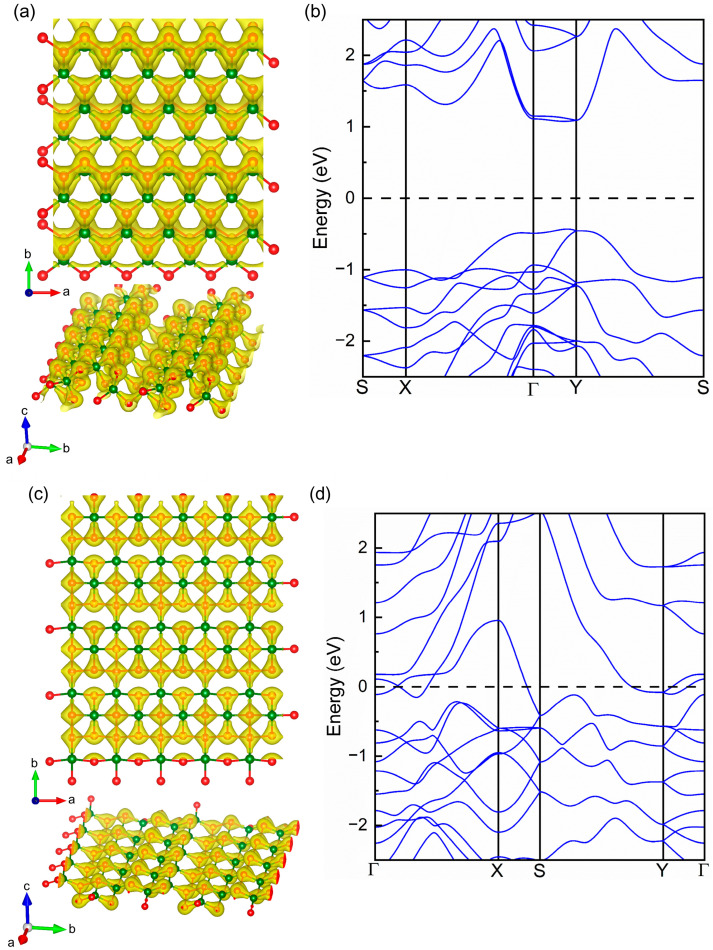
Electron density distributions of a pristine (**a**) and strained (**c**) orthorhombic SiP_2_, and band structures of a pristine (**b**) and strained (**d**) orthorhombic SiP_2_. The isosurfaces of electron density distribution in (**a**,**c**) are separated by 0.07 e/Å^3^. The strain applied in (**c**,**d**) is a biaxial strain of 30.4%. The red and dark green balls represent the P and Si atoms, respectively.

**Figure 8 molecules-28-06514-f008:**
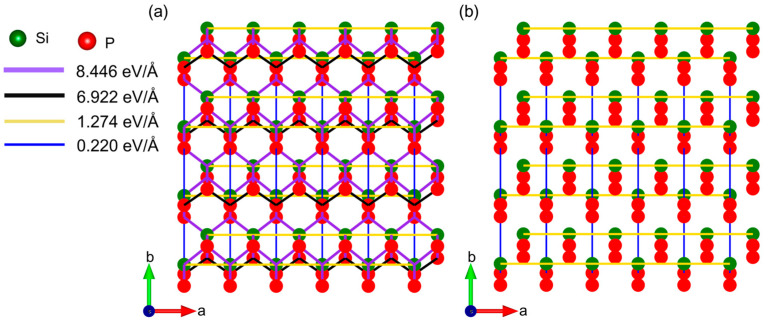
The top views of interatomic force constants of the orthorhombic SiP_2_ monolayer. (**a**) Overview of the larger interatomic force constants. (**b**) View of the largest interatomic force constants along the *a* and *b* directions. The red and dark green balls represent the P and Si atoms, respectively.

**Figure 9 molecules-28-06514-f009:**
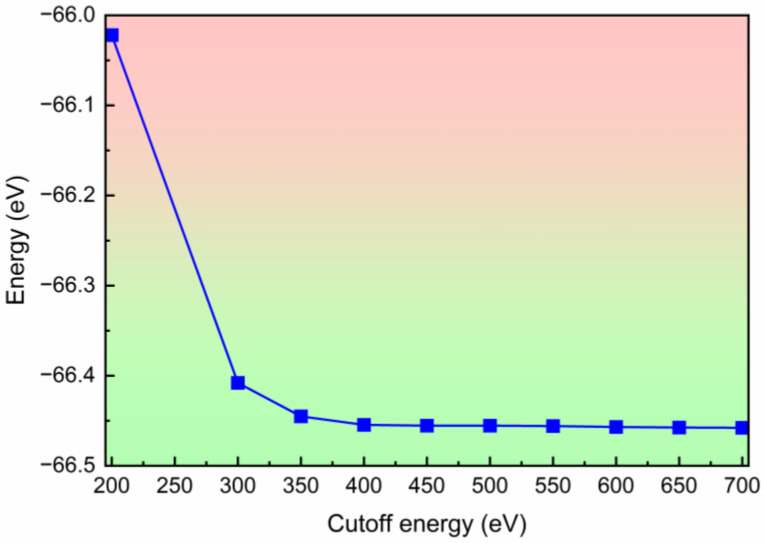
The influence of cutoff energy on the energy of the orthorhombic SiP_2_ monolayer.

**Table 1 molecules-28-06514-t001:** Structure parameters for a pristine orthorhombic SiP_2_ monolayer with and without rain, including lattice constant (a and b), layer thickness, and P-P and P-Si bond lengths.

System	Lattice Constants (Å)		Layer Thickness (Å)	Bond Length (Å)	
	*a*	*b*		P-P	P-Si
0%	3.46	10.28	5.58	2.269	2.281
17%-a	4.06	10.32	5.43	2.369	2.313
20%-b	3.46	12.34	5.41	2.269	2.326
40%-b	3.46	14.39	4.65	2.266	2.362
10.6%-ab	3.83	11.37	4.85	2.369	2.305
24%-ab	4.29	12.75	4.80	2.478	2.297
30.4%-ab	4.51	13.41	2.76	2.441	2.251
35.4%-ab	4.68	13.92	1.79	2.285	2.287

## Data Availability

The data presented in this study are available upon request from the corresponding author.

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
