# Peer review of "Anisotropic Mechanical Properties of Orthorhombic SiP2 Monolayer: A First-Principles Study"

_molecules, 2023, doi:10.3390/molecules28186514_

Round 1

Reviewer 1 Report

The manuscript entitled: “Anisotropic mechanical properties of orthorhombic

SiP2 monolayer: A first-principles study”, by Yinlong Hou, Kai Ren, Yu Wei,

Dan Yang, Zhen Cui and Ke Wang, presented a study of two-dimensional (2D)

SiP2 in orthorhombic phase.

This is an interesting topic of study and generates great interest in the

scientific community due to its wide range of applications, as it contributes

to the understanding of 2D materials and applications in the nanoelectronic

industry. The investigations are well conducted and the article is well written.

I have some suggestions to improve the presentation of the results.

Please, consider the following points.

1. Abstract.

1.1. I suggest that the authors write the sentences of the text in scientific

format. Please improve the sentence:

1.1.1. “...suggesting the infinite potential applications of orthorhombic SiP2

monolayer in…”

2. Introduction.

2.1 Please, consider improving some sentences:

2.1.1 “For a materials...”

2.2. The authors could improve the discussion about anisotropy and the

types of anisotropy.

2.4. In addition to the in-plane anisotropy, is it interesting to study the out-of-

plane anisotropy for this material?

3. Computational Details:

3.1. Suggestion: correct the word “general gradient approximation” by

generalized gradient approximation.

3.2 Why was the 600eV of cutoff considered in all calculations? What were

the PAW projectors used?

4. Results and Discussion.

4.1. Please enlarge the axis and improve the quality of Figure 2.

4.2. I suggest that authors move this sentence below to the introduction of

the article:

4.2.2. Bulk SiP2 has three allotropes including pyrite-type phase,

orthorhombic phase, and tetragonal phase. The pyrite-type phase belongs to

Pa3 (No.205) space group with non-Van der Waals crystal structure[46],

while the orthorhombic and tetragonal phases show 2D layered structure

with Pbam (No.55)[47] and P-421m (No.113)[48-50] space groups,

respectively. In this paper, we focus on the 2D SiP2 monolayer exfoliated

from orthorhombic phase, because the 2D flakes of orthorhombic SiP2 has

been prepared[34].

4.3 Are the lattice parameters optimized independently? Does this relaxation

introduce strain into the system?

4.4. I suggest to the authors that they improve the discussion and clarify the

following sentences:

4.4.1. “...were relaxed in the optimized unit cell unit the energy and...”

4.4.2 “...also suggesting the quasi-one-dimensional mechanical behavior

along a direction in orthorhombic SiP 2 monolayer…”

4.4.3 “... is flattened obviously, resulting...”

4.4.4 “...P-P and P-Si bonds increases obviously with the increase of strain...”

4.5. Does the applied strain of 30.4% make physical sense?

4.6. Please, correct the word “prestine” in the caption of Figure 7 and the

word “quai-one-dimensiona” on page 9.

5. Some references are not abbreviated. Please improve the presentation of

references.

The article is well written, but the scientific writing and English quality can be improved.

Author Response

We thank the reviewer for reading our manuscript carefully and giving the above positive comments. We have revised the manuscript according to the referee’s suggestions, and implemented a proof-reading carefully to eliminate the language issues and types. The changes in manuscript have been highlighted in the revised version. Below is our response to the comments.

Reviewer 2 Report

Dear authors, please see the remarks presented in the attached review document.

Dear authors, please see the remarks presented in the attached review document.

Author Response

We thank the reviewer for reading our manuscript carefully. We have revised the manuscript according to the referee’s suggestions. The changes in manuscript have been highlighted in the revised version. Below is our response to the comments.

Reviewer 3 Report

This manuscript presents the first-principle study on the anisotropic mechanical properties of monolayer orthorhombic SiP2. The authors presented systematic study on the anisotropy in Young's modulus, Poisson's ratio, strain-stress relations and etc. However, the quality of data presentation and discussion must be improved for this manuscript to be considered by Molecules.

1. The overall figure quality of this manuscript needs to be improved. a) The choice of red and green colors for Si and P atoms does not provide very good contrast and sometimes it is difficult to see the geometry and atomic arrangement. b) Among all atomic structures, only figure 1 shows legend for Si and P atoms. c) Figure 4 should have a-axis label along 0 degree, as well as description of the orientation in the caption. d) The interatomic force constant in figure 8 is very unclear. The width differences for different lines are not noticable. Too many lines overlapping within one structure and no emphasis. I would suggust the authors to have one zoom-in atomic structure with labeling for each atomic force constant, and one overview atomic structure showing the difference within a-b plane.

2. The introduction part is not well structured. The emphasis of the first paragraph should on anisotropy for 2D materials and why researchers care this phenomenon. But the authors spend half of the paragraph discussing broken inversion symmetry in MoS2 and twisted bilayer graphene, which are not directly related to the topic. There is also no transition for the anisotropy discussion. For the second paragraph, the authors simply put together all the literature related to 2D SiP2, without any key point mentioned. The authors should present the high level summary on the crystal structure and materials properties of 2D SiP2 first, before diving into specific literatures on what has been done in terms of anisotropy study and what is missing.

3. For the anisotropic Young's modulus and Poisson's ratio, the authers should at least provide derivation of equation (1) and (2) in supporting information, or cite references for the equation. 

4. The negative poisson's ratio for specific orientation is interesting, but the negative portion is not clearly labeled in Figure 4b. The authors should reconsider how to present this finding clearly.

5. I am not convinced by the interatomic force constant discussion to explain the origin of mechenical anisotropy. Why comparing a-axis Si-Si force constant with b-axis P-P force constant? Do other atomic bonds not have impact on a- and b-axis? The authors needs to provide better explanation on this part.

Overall English quality is fine. But I still recommend the authors to work on transitioning between different topics and clearly stating the motivation for specific discussion.

Author Response

(The authors gave the same response as above.)

Round 2

Reviewer 2 Report

Dear authors, I see major improvements of your manuscript. You answered clearly and satisfactory to all my review remarks. I recommend to publish the manuscript in the MDPI journal.

Reviewer 3 Report

For the revised manuscript, the authors have addressed most of the review comments. I agree to accept the manuscript in present form.

The English language is fine.